# Cerebellar Golgi cell models predict dendritic processing and mechanisms of synaptic plasticity

**Stefano Masoli**[1], **Alessandra Ottaviani**[1], **Stefano Casali**[1], **Egidio D'Angelo**[1,2]*

**1** Department of Brain and Behavioral Sciences, University of Pavia, Pavia, Italy, **2** Brain Connectivity Center, IRCCS Mondino Foundation, Pavia, Italy

* dangelo@unipv.it

**Data Availability Statement:** The models are available on the Brain Simulation Platform (BSP) of the Human Brain Project (HBP) as a "live paper" containing a selection of routines and optimization scripts and their codebase is uploaded on

## Abstract

The Golgi cells are the main inhibitory interneurons of the cerebellar granular layer. Although recent works have highlighted the complexity of their dendritic organization and synaptic inputs, the mechanisms through which these neurons integrate complex input patterns remained unknown. Here we have used 8 detailed morphological reconstructions to develop multicompartmental models of Golgi cells, in which Na, Ca, and K channels were distributed along dendrites, soma, axonal initial segment and axon. The models faithfully reproduced a rich pattern of electrophysiological and pharmacological properties and predicted the operating mechanisms of these neurons. Basal dendrites turned out to be more tightly electrically coupled to the axon initial segment than apical dendrites. During synaptic transmission, parallel fibers caused slow Ca-dependent depolarizations in apical dendrites that boosted the axon initial segment encoder and Na-spike backpropagation into basal dendrites, while inhibitory synapses effectively shunted backpropagating currents. This oriented dendritic processing set up a coincidence detector controlling voltage-dependent NMDA receptor unblock in basal dendrites, which, by regulating local calcium influx, may provide the basis for spike-timing dependent plasticity anticipated by theory.

## Author summary

The Golgi cells are the main inhibitory interneurons of the cerebellum granular layer and play a fundamental role in controlling cerebellar processing. However, it was unclear how spikes are processed in the dendrites by specific sets of ionic channels and how they might contribute to integrate synaptic inputs and plasticity. Here we have developed detailed multicompartmental models of Golgi cells that faithfully reproduced a large set of experimental findings and revealed the nature of signal interchange between dendrites and axosomatic compartments. A main prediction of the models is that synaptic activation of apical dendrites can effectively trigger spike generation in the axonal initial segment followed by rapid spike backpropagation into basal dendrites. Here, incoming mossy fiber inputs and backpropagating spikes regulate the voltage-dependent unblock of NMDA channels and the induction of spike timing-dependent plasticity (STDP). STDP, which was

ModelDB: https://senselab.med.yale.edu/modeldb/ShowModel?model=266806#tabs-1.

**Funding:** This research was supported by the European Union's Horizon 2020 Framework Programme for Research and Innovation under the Specific Grant Agreement No. 785907 (Human Brain Project SGA2) to ED. This research was also supported by the MNL Project "Local Neuronal Microcircuits" of the Centro Fermi (Rome, Italy) to ED. Model optimizations and simulations were performed on the Piz Daint supercomputer (CSCS – Lugano) with a specific grant (special proposal 03) to ED&SM and using computing resources provided through the PRACE Project 2018184373 to ED&SM. The funders had no role in study design, data collection and analysis, decision to publish, or preparation of the manuscript.

**Competing interests:** The authors have declared that no competing interests exist.

predicted by theory, may therefore be controlled by contextual information provided by parallel fibers and integrated in apical dendrites, supporting the view that spike timing is fundamental to control synaptic plasticity at the cerebellar input stage.

## Introduction

The cerebellar Golgi cell [1–3] is the principal inhibitory interneuron of the cerebellar granular layer [4] and is supposed to play a critical role for spatio-temporal reconfiguration of incoming inputs by regulating neurotransmission and synaptic plasticity along the mossy fiber–granule cell pathway [5–7]. Golgi cells are wired in feed-forward and feed-back loops with granule cells and form a functional syncytium through gap junctions [8]. The excitatory inputs are conveyed by mossy fibers and granule cell ascending axons on the basal dendrites and by parallel fibers on the apical dendrites [9]. The inhibitory inputs come from other Golgi cells [10] and from Lugaro cells [11,12]. The Golgi cells eventually inhibit large fields of granule cells through an extended axonal plexus. Fundamental issues that remain unexplored are how synaptic inputs control Golgi cell spike generation and whether dendritic processing provides the basis for spike-timing dependent plasticity (STDP), which has been predicted by theory [13]. Interestingly, mossy fiber–Golgi cell synapses express NMDA channels (fundamental for synaptic plasticity) at mossy fiber synapses [9] and the dendrites express a diversified set of Ca, Na and K ionic channels [14] that could impact on dendritic computation. A prediction about the possible interactions of these multiple active properties is quite hard and requires a detailed computational analysis of the electrogenic architecture of the neuron and of synaptic integration.

The complement of Golgi cell ionic channels was initially determined with somatic whole-cell recordings using bath-applied pharmacological blockers [15]. In the absence of more detailed information, an initial computational model assumed that Golgi cell intrinsic electro-responsiveness could be generated by active properties concentrated in the soma, while a passive dendritic tree was used to balance the electrotonic load[16,17]. That same model, once reimplemented into a realistic morphology, was used to test electrical transmission through gap junctions in Golgi cell pairs [8]. But since then, the discovery of active dendritic properties [14] and of the complexity of Golgi cell wiring in the local microcircuit [9,10] has changed the perspective. In order to get insight on how this neuron might work based on the interaction of its membrane and synaptic mechanisms, we have developed realistic computational models of Golgi cells based on detailed morphological reconstructions and on the new electrophysiological and immunohistochemical data that have become available about the properties of dendrites and synapses. The new models, further than faithfully reproducing the rich pattern of Golgi cell electrophysiological responses recorded *in vitro* and *in vivo*, provided testable predictions for dendritic processing, synaptic integration and STDP. Interestingly, NMDA receptor-dependent STDP in basal dendrites turned out to be controlled by information provided by parallel fibers and integrated in apical dendrites, suggesting the way spike timing might supervise synaptic plasticity at the cerebellar input stage.

## Results

We have reconstructed, optimized and simulated 8 multicompartmental GoC models based on detailed morphologies obtained from mouse cerebellum (Figs 1A and S1). The voltage dependent ionic channels were distributed over the entire neuronal structure, based on indications derived from electrophysiology and immunostaining [14], generating models with active

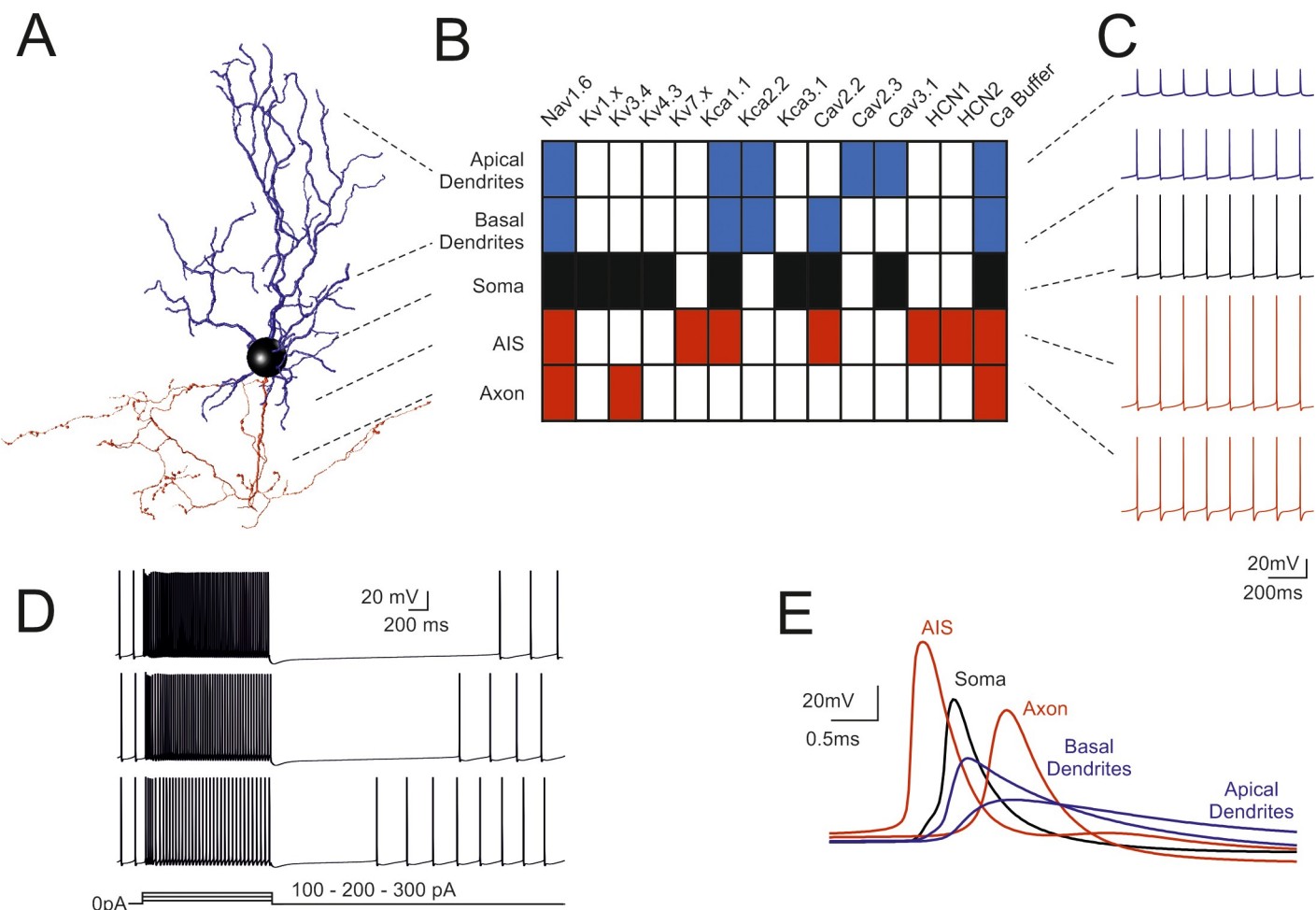

**Fig 1. Fundamental electroresponsiveness in a multicompartmental Golgi cell model.** (A) 3D morphological reconstruction of a GoC [from [19]] including dendrites (blue), soma (black), axonal initial segment and axon (red). The apical dendrites where distinguished based on their projection into the molecular layer (ML), while basal dendrites remained in the granular layer (GL). (B) Localization of ionic channels in the different model sections according to immunohistochemical, electrophysiological and pharmacological data. (C) Spikes during pacemaking in the different sectors indicated in B. Note the strong spike reduction in apical GoC dendrites but the effective backpropagation into basal dendrites. (D) Responses to 100-200-300 pA step current injections in the model. (E) Spikes (taken from C) on expanded time scale. Note that spikes are generated in the AIS and then propagate with increasing delay to soma, axon, basal and apical dendrites. In this latter, spikelets are severely delayed, reduced and slowed down. When not otherwise specified, examples in the next figures will be taken from this same Golgi cell model.

properties in dendrites, soma, axonal initial segment (AIS) and axon. The ionic channel sub-types, their distribution over the compartments belonging to the same section (Fig 1B), their maximum conductance ranges, as well as the spike features used as template, were the same for the 8 Golgi cell models. The models were tuned through automatic optimization of maximum conductance values based on a genetic algorithm, while targeting an electrophysiological template (see Materials and Methods), and were carefully validated against a wealth of experimental data. All the models were autorhythmic (Fig 2C) and current injection (0.1nA to 0.5nA) increased the firing rate (Fig 2D). During the current steps, spike discharge showed frequency adaptation. At the end of the steps, membrane potential showed a hyperpolarization followed by a pause in spike firing. These properties resembled those observed experimentally [15].

The action potentials were generated in the AIS with forward propagation in the axon and back-propagation in the soma and dendrites (Fig 1E). Axonal invasion took about 1 ms

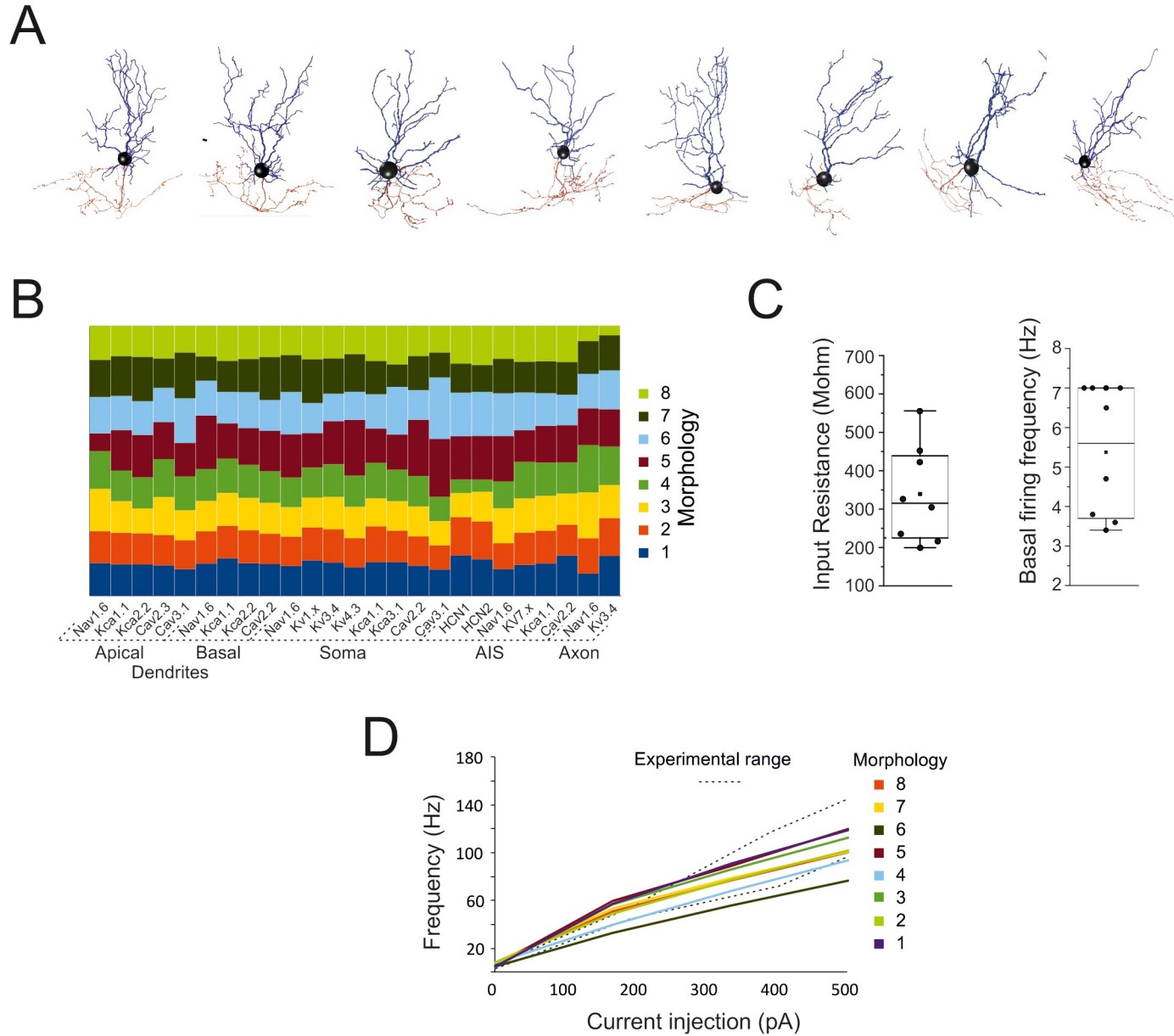

**Fig 2. Quantitative aspects of Golgi cell models.** (A) The morphologies of the 8 Golgi cells used for modeling. (B) The table shows the balance of maximum ionic channel conductances in the 8 GoC models, demonstrating their similarity. (C) The box-and-whiskers plots show the distribution of input resistance (measured from current transients elicited by 100 ms steps from -70m to– 80 mV in the soma) and basal firing frequency. (D) Frequency-intensity curves for the 8 GoC models compared to the experimental range [dotted lines from [15]]. The same color code is used for the cells in B and D.

according to experimental delay times in the Golgi cell—granule cell circuit [18], while dendritic back-propagation occurred in about 0.5 ms. Backpropagating spikes were 50–80% of AIS size in the basal dendrites but were severely reduced to 10–20% of AIS size in the apical dendrites, as expected from the differential distribution of Na channels anticipated by [14].

## Intrinsic model electroresponsiveness with somatic current injection

Following parameter optimization and validation, all the models behaved as typical Golgi cells and showed minor differences in the maximum ionic conductance values (Fig 2A and 2B),

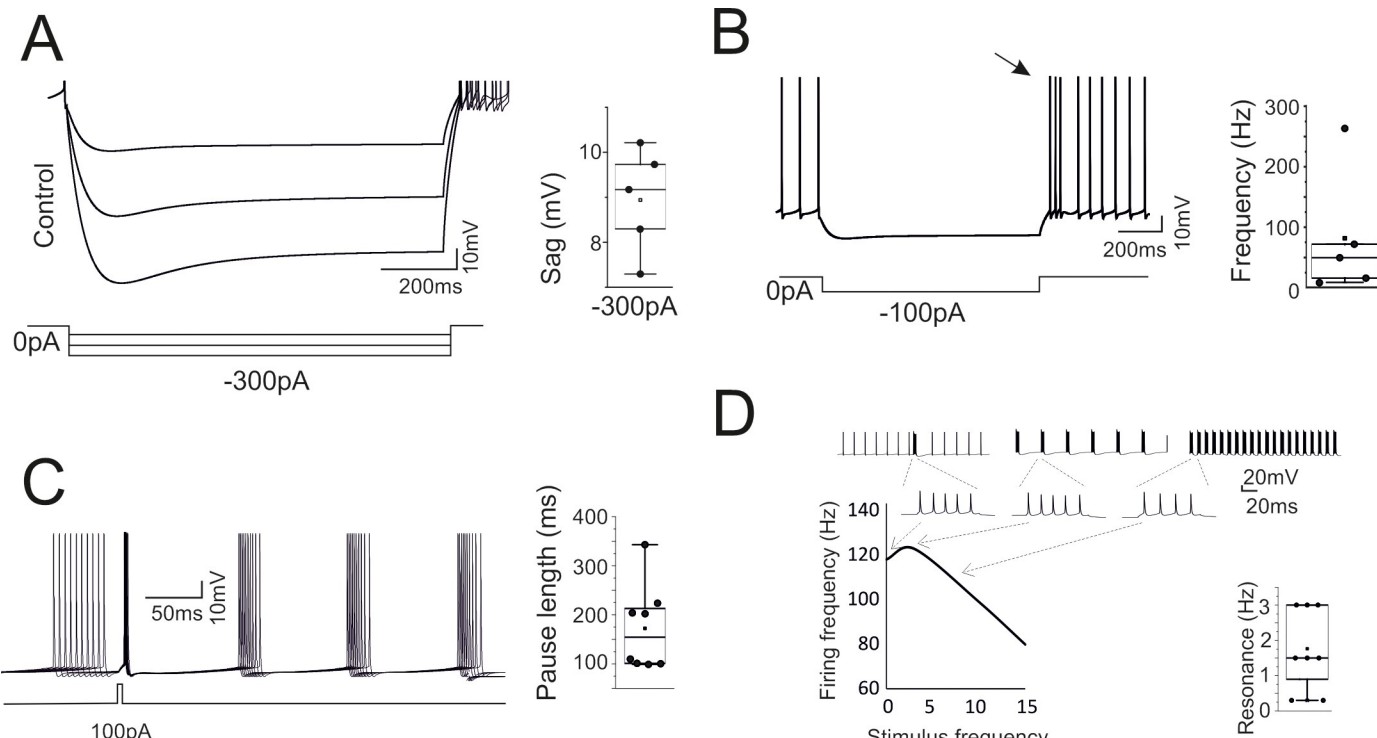

**Fig 3. High level model validation: sags, rebounds, phase-reset and resonance.** The model was simulated in operating conditions capable of revealing typical GoC behaviors not considered for model construction. (A) Response to hyperpolarizing currents. The model generates sagging hyperpolarization. The box-and-whiskers plot shows sag amplitude in the 8 models. (B) Return to resting activity following hyperpolarizing currents. The model transiently generates a rebound burst (arrow). The box-and-whiskers plot shows burst frequency in the 8 models. (C) Responses following a depolarizing current pulse. The model shows phase-reset. The box-and-whiskers plot shows the pause length in the 8 models. (D) Responses following injection of current steps at different frequencies. The model shows resonance of output spike frequency. The traces illustrate the resonance protocol and an enlargement of the spike bursts used for measuring the output frequency. The box-and-whiskers plot shows resonance frequency in the 8 models.

demonstrating that individual cell morphology did not impact remarkably on physiological properties (S2 Fig).

In the Golgi cell models, the average input resistance was 339.3 ± 128.3 MΩ (n = 8 models) and spontaneous firing frequency was at 5.4±1.7 Hz (n = 8 models) (Fig 2C). The frequency / current relationship was almost linear in a range up to ~100 Hz (Fig 2D).

While the properties reported in Fig 2 were counter checked during the optimization process (the models that did not comply with the reference parameters were eliminated), a more extended set of electroresponsive properties typical of the Golgi cells [15,20] emerged providing an important proof for model validation. A *sag* of 5–8 mV was elicited during negative current injection (Fig 3A). *Rebound bursting* was elicited on return to rest from a hyperpolarized potential (Fig 3B). *Phase-reset* occurred following a brief current pulse injection (Fig 3C). *Resonance* emerged by challenging the model with repetitive current steps (repetition frequency 1–15 Hz), which caused a preferential response at low frequency (1–4 Hz) (Fig 3D). that were previously remapped onto specific ionic channels [16,17]: phase reset was related to calcium entry and the subsequent activation of Kca2.2 channels, the length of the interspike interval was modulated by the H current, rebound bursts were generated by low-threshold Ca channels, resonance depended on Kv7.x. Therefore, the information extracted from the template was sufficient to optimize the ionic conductance balance toward properties beyond those directly assessed during model construction.

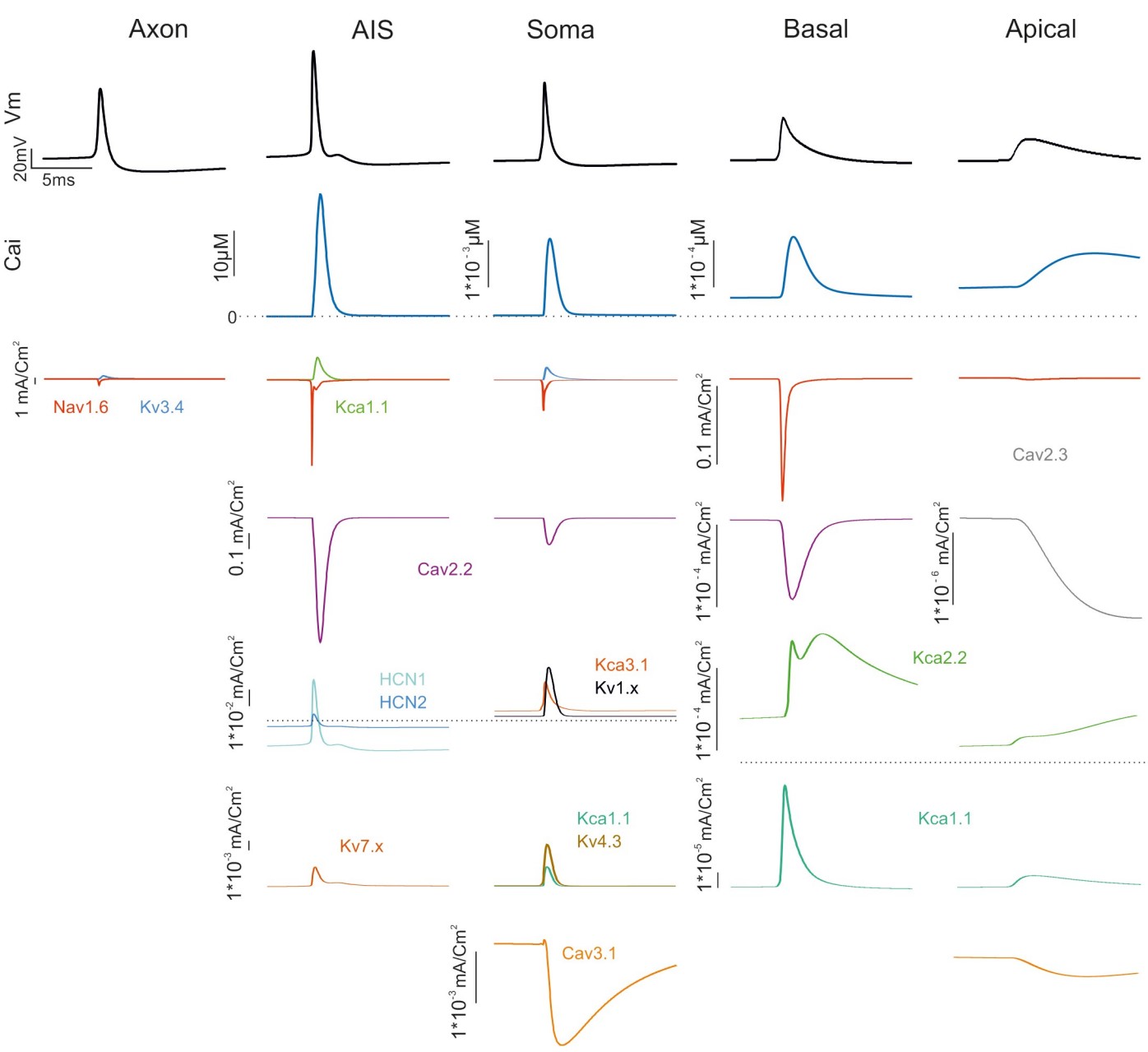

**Fig 4. Ionic currents in model sections.** The figure shows the ionic currents and calcium concentration changes generated by membrane channels in the GoC model when a spike occurs during autorhythmic firing. Note the localization of channels in different sections and the different calibration scales.

### Ionic currents in different model compartments

The ionic currents generated in the different Golgi cell model compartments are illustrated in Fig 4. While the distribution of ionic channels was constrained by immunohistochemistry, electrophysiology and pharmacology, the maximum ionic conductances were the true unknown of the models and were set through automatic parameter optimization. These values eventually determined the ionic current densities shown in Fig 4.

*AIS and soma.* The AIS, in addition to sodium channels (Nav1.6), also hosted high-threshold calcium channels, potassium channels and H channels (Cav2.2, Kca1.1, HCN1/ HCN2,

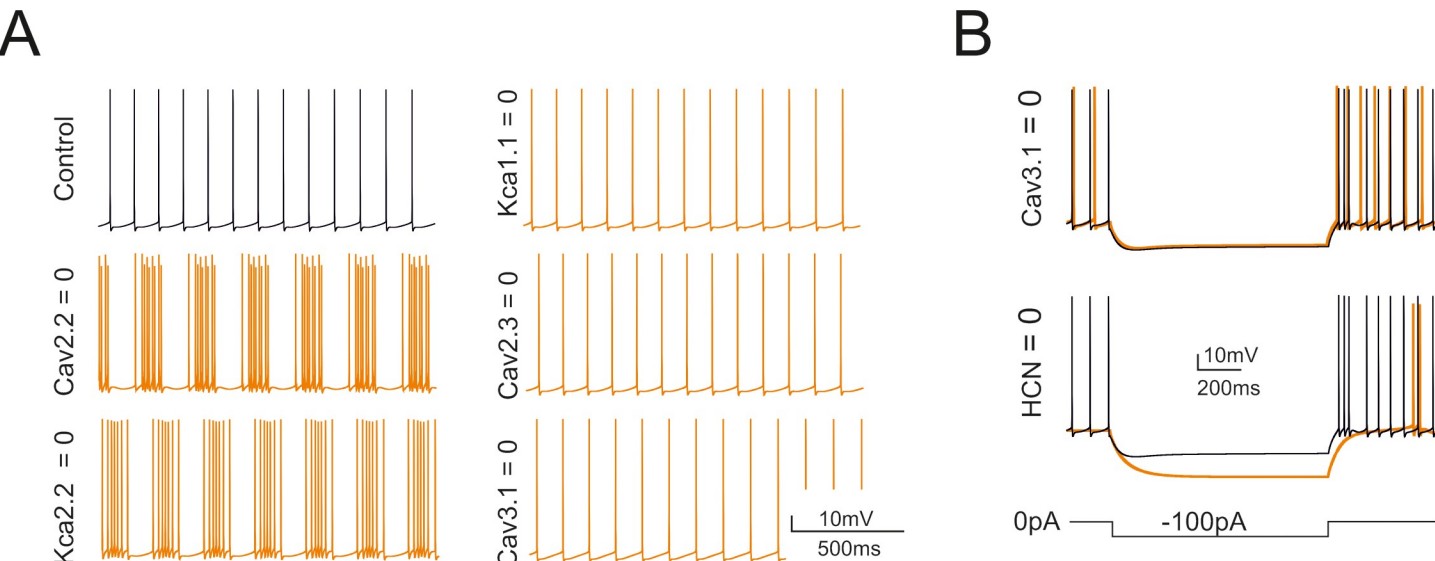

**Fig 5. Selective switch-off of ionic mechanisms.** Counter-testing of the GoC model obtained by switch-off of ionic channels in specific model sections. (A) Effect on basal firing of the switch-off of Cav2.2, Kca2.2, Kca1.1, Cav2.3, Cav3.1. These simulations imitate the experiments on pharmacological block reported in [14]. (B) Effect on the responses to negative currents steps (during the step and in the rebound phase) of the switch-off of HCN (HCN1 and HCN2) and Cav3.1.

Kv7.x). The soma hosted sodium channels (Nav1.6), high-threshold and low-threshold calcium channels (Cav2.2, Cav3.1) and various potassium channels (Kca1.x, Kca3.1, Kv3.4). The AIS Na current proved critical for action potential generation [21] and had a 3 times higher density compared to soma (13.5 mA/cm$^2$ vs. 4.5 mA/cm$^2$). The Cav2.2 channels were also expressed in AIS and caused the intracellular calcium concentration $[Ca^{2+}]_i$ to reach levels 3 times higher than in the soma. This peculiar phenomenon was also observed in Cartwheel neurons of the Cochlear Nucleus and in cerebral cortex L5 Pyramidal cells [22,23]. The main effect of the calcium current and of the associated $[Ca^{2+}]_i$ increase was to activate Kca1.1 and Kca3.1 and to regulate action potential generation along with the voltage-dependent K current Kv3.4. HCN1 and HCN2 increased the action potential threshold and regulated the pacemaker cycle, while Kv7.x was critical for resonance as anticipated by [16,17].

*Dendrites*. The basal and apical dendrites hosted, in addition to a low density of sodium channels (Nav1.6), a repertoire of calcium channels (Cav2.2, Cav2.3, Cav3.1) and calcium-dependent potassium channels (Kca2.2). These currents sustained spike backpropagation, which was much more effective in the basal than apical dendrites.

### Selective switch-off of ionic currents in the model

The impact of ionic channels on spontaneous firing was evaluated by selective switch-off imitating pharmacological blockade [14,15] (Fig 5). The switch-off of the main calcium current (Cav2.2) uncovered oscillatory bursting at 4-5Hz. A similar oscillatory bursting was seen with the switch-off of Kca2.2 in apical and basal dendrites, indicating that the main effect of Cav2.2 was to control Kca2.2. The switch-off of Cav2.3 had no visible impact on spontaneous firing. The switch-off of Kca1.1 and Cav3.1 caused just a slight increase in spontaneous firing frequency. As a whole, during spontaneous activity, the apical dendrites ionic channels showed little involvement. The switch-off of HCN1 and HCN2 abolished the sag and reduced spontaneous firing frequency to 2.2 ±2.1 Hz (n = 8). The switch-off of Cav3.1 in the apical and the somatic sections abolished rebound bursts. All these tests were in good matching with

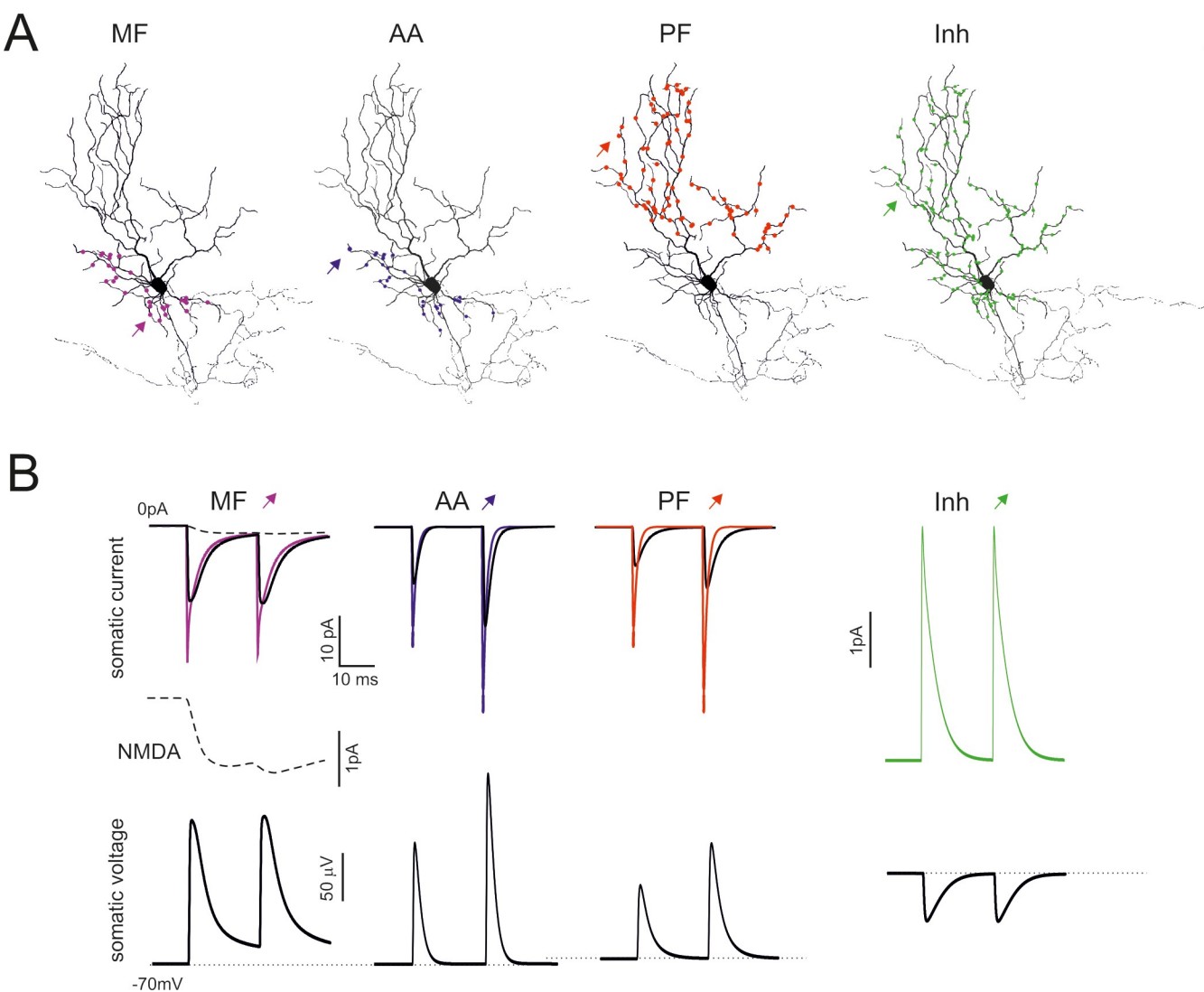

**Fig 6. Model responses to synaptic inputs.** (A) The figure shows the localization of mossy fibers (MF), ascending axon (AA), parallel fiber (PF), and inhibitory (Inh) synapses. (B) Synaptic currents generated by single MF, AA, PF and Inh synapses. The thin line is the current generated at the synaptic site, the thin line is the current recorded in the soma. The dashed line is the NMDA current. The corresponding EPSPs and IPSPs are reported at the bottom. In all cases the responses are elicited by a stimulus pair at 50 Hz.

pharmacological experiments [14,15] and provided a strong support to model validation by showing that the ionic maximum conductances set through automatic optimization were mechanistically related to the main functional properties of the Golgi cell.

## Synaptic properties of the Golgi cell model

The implications of dendritic processing were analyzed by using realistic representations of synaptic transmission, including presynaptic release dynamics of vesicle cycling, neurotransmitter diffusion and postsynaptic receptors activation (see [24–28]. The models were endowed with parallel fiber, ascending axon and mossy fiber synapses using information taken from literature [9] (Fig 6A). The parallel fiber on apical dendrites and ascending axon on apical and basal dendrites were facilitating (paired pulse ratio, PPR>1) and were endowed with AMPA

receptors. The mossy fiber on the basal dendrites, at least 20μm away from the soma, were neither facilitating nor depressing (paired pulse ratio, PPR = 1) and were endowed with AMPA receptors and NMDA NR2B receptors. The basal dendrites received GABA-A receptor-mediated inputs (Fig 6A). The only part of the neuron devoid of synapses was the dendritic trunk hiding in the PC layer.

Unitary EPSCs at their generation site had fast kinetics but the EPSCs were reduced in amplitude and slowed down when seen from the soma, as expected from electrotonic decay along the dendrites (Fig 6B). The EPSCs generated on apical dendrites were more heavily filtered than those generated on basal dendrites. Moreover, mossy fiber EPSCs had a small but sizeable NMDA current component. In aggregate, the simulations confirmed the interpretation of experimental EPSC properties [9]. Unitary IPSCs were also similar to those reported experimentally [10]. The unitary EPSCs and IPSCs had small size (10 pA range at -70 mV) and generated comparatively small voltage deflections with PPR resembling that of the EPSCs and IPSCs (Fig 6B). The ascending axon EPSPs were faster than parallel fiber EPSPs but slower that mossy fiber EPSP, whose kinetics were protracted by the NMDA current.

## The efficiency of synaptic activation

While the picture shown in Fig 7B is a direct reflection of the known EPSC properties and dendritic filtering, the synaptic response is expected to change when the number of synapses increases and EPSPs grow enough to engage dendritic voltage-dependent mechanisms (Fig 8A). In order to predict the efficiency of spatial integration over the different input channels, the model was stimulated with an increasing number of synchronous synaptic inputs. The mossy fiber, ascending axon and parallel fiber inputs required the simultaneous activation of 2–3 synapses to generate a spike, supporting the ability of the GoC models to integrate synaptic inputs with high efficiency.

When the model was stimulated with a short burst (5 spikes@100Hz) delivered either through the mossy fibers (20 synapses), the ascending axon (20 synapses), the parallel fibers (89 synapses), all the three excitatory inputs generated a short spike burst followed by a pause that effectively reset the pacemaker cycle (cf. the PSTHs in Figs 7B and 3C) (S1–S3 Videos). Synaptic inhibition delivered in correspondence with the excitatory burst (20 synapses) reduced the number of emitted spikes but did not impact remarkably on the pause length (Fig 7C). The effect of inhibition on the number of emitted spikes and of the subsequent pause depended on the number of excitatory synapse and was poorly sensitive to the relative excitation/inhibition phase in the ±10 ms range (Fig 7D). These response patterns resembled those observed *in vivo* following punctuate facial stimulation [29].

## Active dendritic currents during synaptic transmission

The results reported above suggest that dendritic excitation in Golgi cells is rather complex. First, given the differential density of Na channels, the basal dendrites show a more efficient spike backpropagation than the apical dendrites (Fig 1C and 1D). Secondly, Ca channels are localized differentially (Fig 1B), with high-threshold Ca channels (Cav2.2) located in basal dendrites (as well as in soma and initial segment) and low-threshold Ca channels (Cav3.1) located in distal dendrites (as well as in soma). We thought these factors could reverberate into an asymmetric dendritic excitation during synaptic transmission.

Fig 8 illustrates the underlying ionic mechanisms and Ca transients. Local spikes in basal dendrites were associated with rapid calcium transients, which were sustained by Cav2.2 and traversed the μM range. Excitation propagating to apical dendrites was slowly integrated by Cav3.1 activation causing Ca spikelets raiding a prolonged calcium wave. The slow local

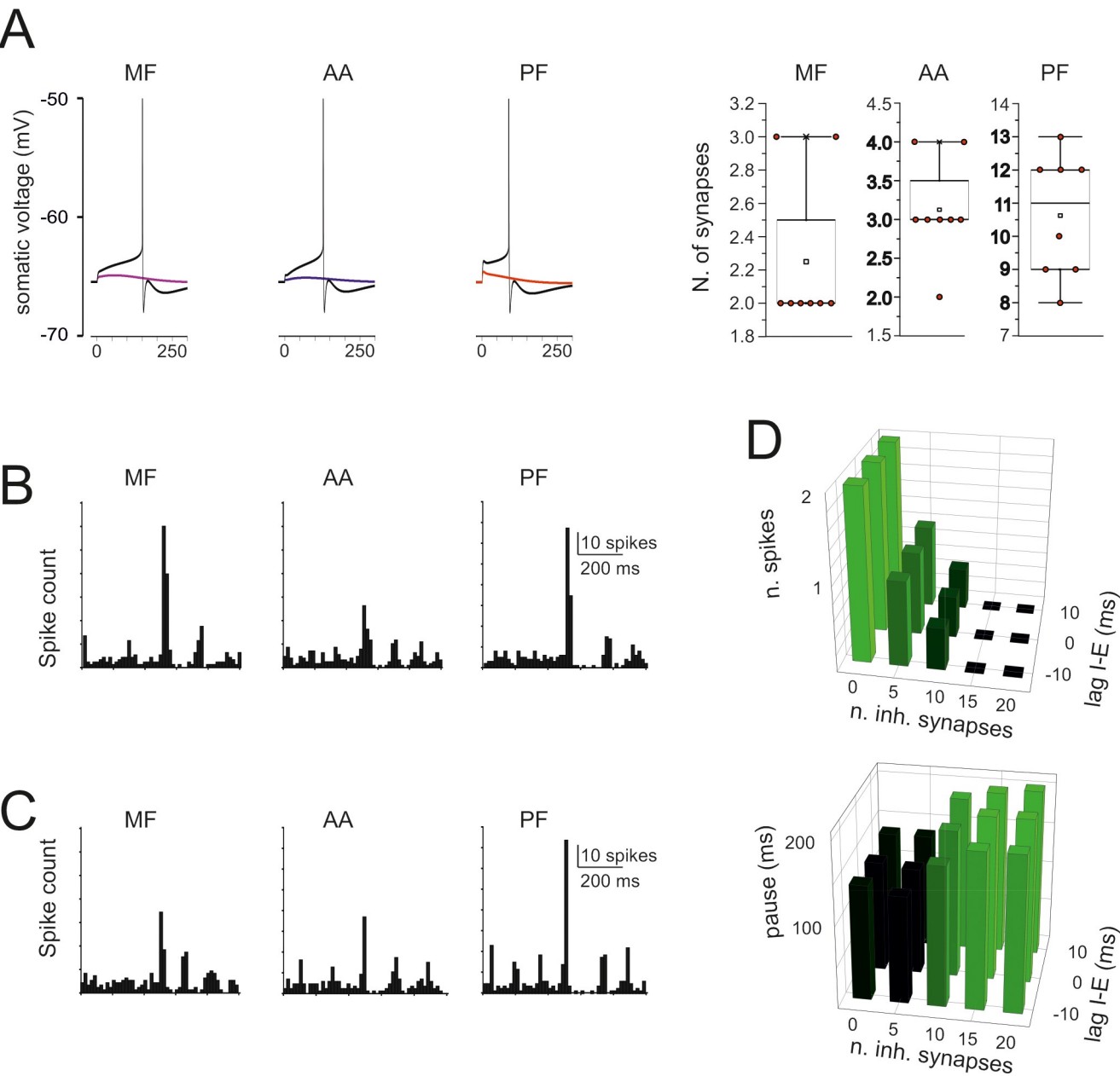

**Fig 7. Simulations of synaptic excitation and inhibition.** (A) The traces show spikes elicited by a single stimulus on 2 MF, 3 AA, and 12 PFs. The number of synapses that need to the activated simultaneously in order to elicit a spike is shown in the box-and-whiskers plots, which report the effect of stimuli delivered after switch-off of pacemaking with a small negative current. (B) PSTH of responses elicited by a short train (5 impulses @100 Hz) to 20 MF, 20 AA, and 89 PF synapses. Note that the response bursts are followed by a pause (corresponding to phase-reset). (C) PSTH of responses elicited by a short train (5 impulses @100 Hz) to 20 MF, 20 AA, and 89 PF synapses plus 20 inhibitory synapses activated 10 ms after the excitatory input. Note that the decrease of response bursts compared to B. (D) Number of spikes and pause length as a function of the number of inhibitory synapse and of the lag of inhibition with respect to excitation.

depolarization in apical dendrites was associated with a massive calcium wave sustained by Cav3.1 and traversing the μM range. The major repolarizing role was sustained by small-K channels (KCa2.2) in apical dendrites, which outperformed big-K channels (KCa1.1) by almost one order of magnitude. These simulations support the conclusion that dendritic processing in Golgi cells is markedly asymmetric.

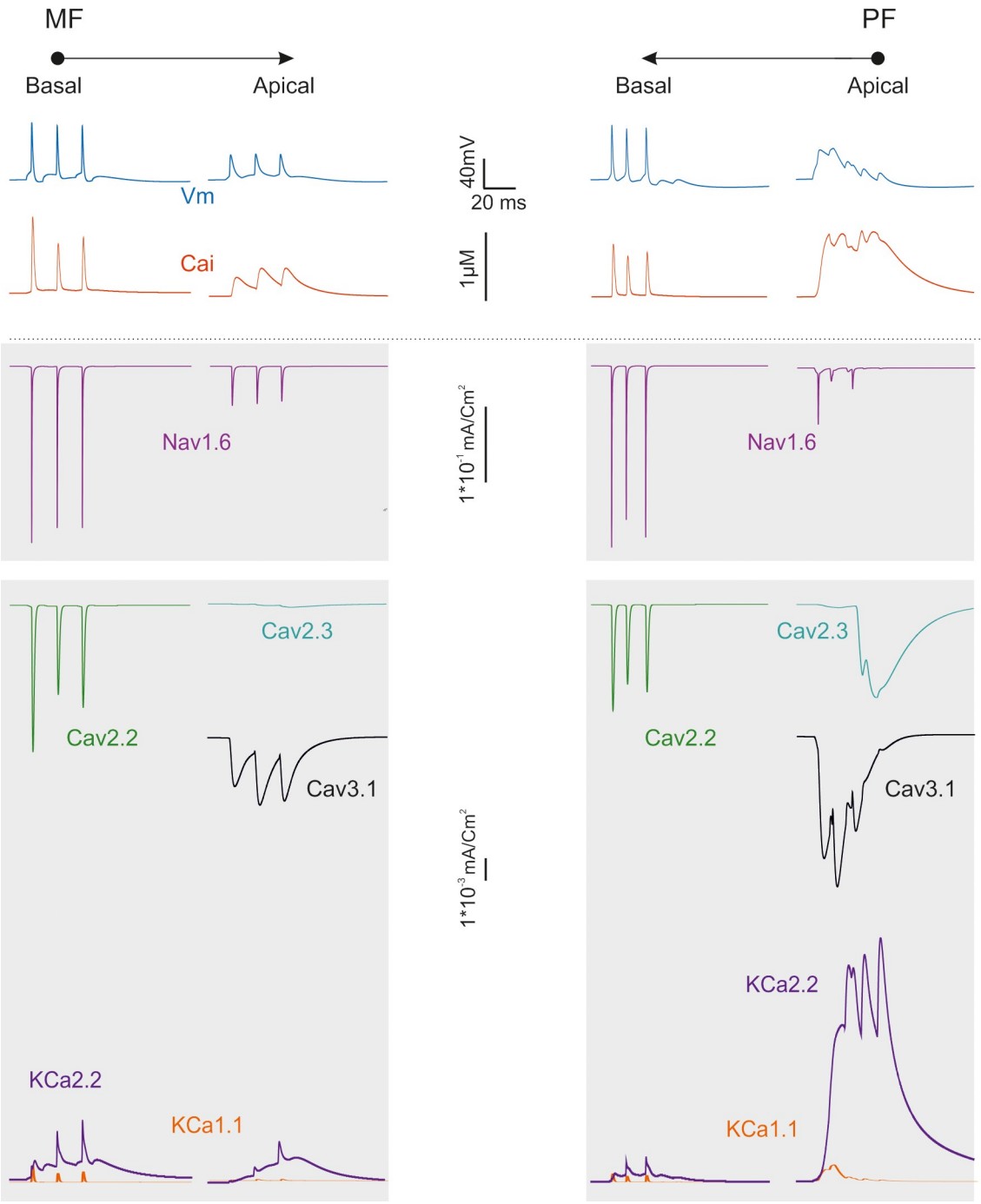

**Fig 8. Ionic current in the dendrites during synaptic transmission.** The traces show the ionic currents and intracellular calcium concentration changes elicited by activation of a short bursts of synaptic activity in either MFs in basal dendrites or PFs in apical dendrites.

## Prediction of coincidence detection and STDP between parallel fiber and mossy fiber activity

Golgi cell dendritic processing would be much enriched by considering the interaction of inputs occurring on basal and apical dendrites (an analogous case has been investigated in

pyramidal neurons; [30,31]. Given the complexity of the synaptic input-output (I/O) space (e.g. see S4 Fig), the interaction between basal and apical dendrites is illustrated for the case of short bursts (5 spikes@100Hz) delivered either through mossy fibers (20 synapses), parallel fibers (89 synapses) or both (Fig 9A). Mossy fiber stimulation on basal dendrites determined rapid synaptic currents that were transmitted to AIS. AIS generated spikes, which travelled back to the basal dendrites and partially to the apical dendrites. Parallel fiber stimulation on apical dendrites activated slow local currents that were transmitted to AIS generating spikes, which travelled back to the basal dendrites and at a much lesser extent to the apical dendrites. Joint mossy fiber and parallel fiber stimulation improved AIS activation generating again a robust backpropagation into basal dendrites. Finally, inhibition severely impaired both spike generation and dendritic backpropagation.

The electrogenic architecture of GoCs revealed by simulations suggested that dendritic processing could implement a coincidence detector driving spike-timing dependent plasticity (STDP) [32,33]. Fig 9B shows the effect of phase-correlated parallel fiber and mossy fiber inputs. When a parallel fiber input precedes a mossy fiber input, this latter falls in the AHP region of the backpropagating spike. The NMDA channels do not unblock (Fig 9C) and the associated Ca influx is small. This would result in LTD. When a parallel fiber input follows a mossy fiber input, this latter intercepts the upstroke of the backpropagating spike. NMDA channels unblock (Fig 9C) raising the associated Ca influx considerably. This would result in LTP. Activation of inhibitory synapses and reduced the burst and its backpropagation, thereby effectively preventing NMDA channel unblock and LTP. A schematic of the possible mechanisms involved and a prediction of STDP with different synaptic input patterns is shown in S4 Fig.

## Discussion

The simulation of multicompartmental models of cerebellar Golgi cells showed an asymmetric communication between cell compartments, with apical dendrites being favored for integration of multiple synaptic inputs and basal dendrites for rapid spike back-propagation and NMDA receptor voltage-dependent regulation. The electrogenic architecture of Golgi cells therefore sets up a cellular coincidence detector of activities conveyed by mossy fibers and parallel fibers, with important reflections on the way Golgi cells perform local computations and regulate STDP.

The multicompartmental models used here allowed to integrate complex data sets and to predict electroresponsive dynamics in functional conditions that would be hard to assess experimentally. The Golgi cell models were parameterized using objective optimization routines targeting cell firing as a template and yielding the maximum conductance of voltage-dependent ionic channels in the soma, dendrites, AIS and axon [14]. The models were validated toward high-level electroresponsive properties including rebound sags and bursts, phase reset and resonance [14,15,20]. Interestingly, all the 8 optimized models reproduced response patterns consistent with experimental Golgi cell activity with a similar ionic conductance balance (cf. Fig 1D) that can be taken as a canonical property of these neurons. But then an obvious question arises: why previous models, with simplified morphology, passive dendrites and a limited set of ionic mechanisms collapsed into the soma [8,16,17,35] were also able to capture the same response patterns? Simpler models already predicted that phase reset was due to calcium entry and the subsequent activation of Ca channels, that the length of the interspike interval was modulated by HCN1 channels, that rebound bursts were generated by low-threshold Ca channels, and that resonance depended on an M-like current. The answer is that all these ionic channels are located in the AIS or in its proximity and do not engage specific

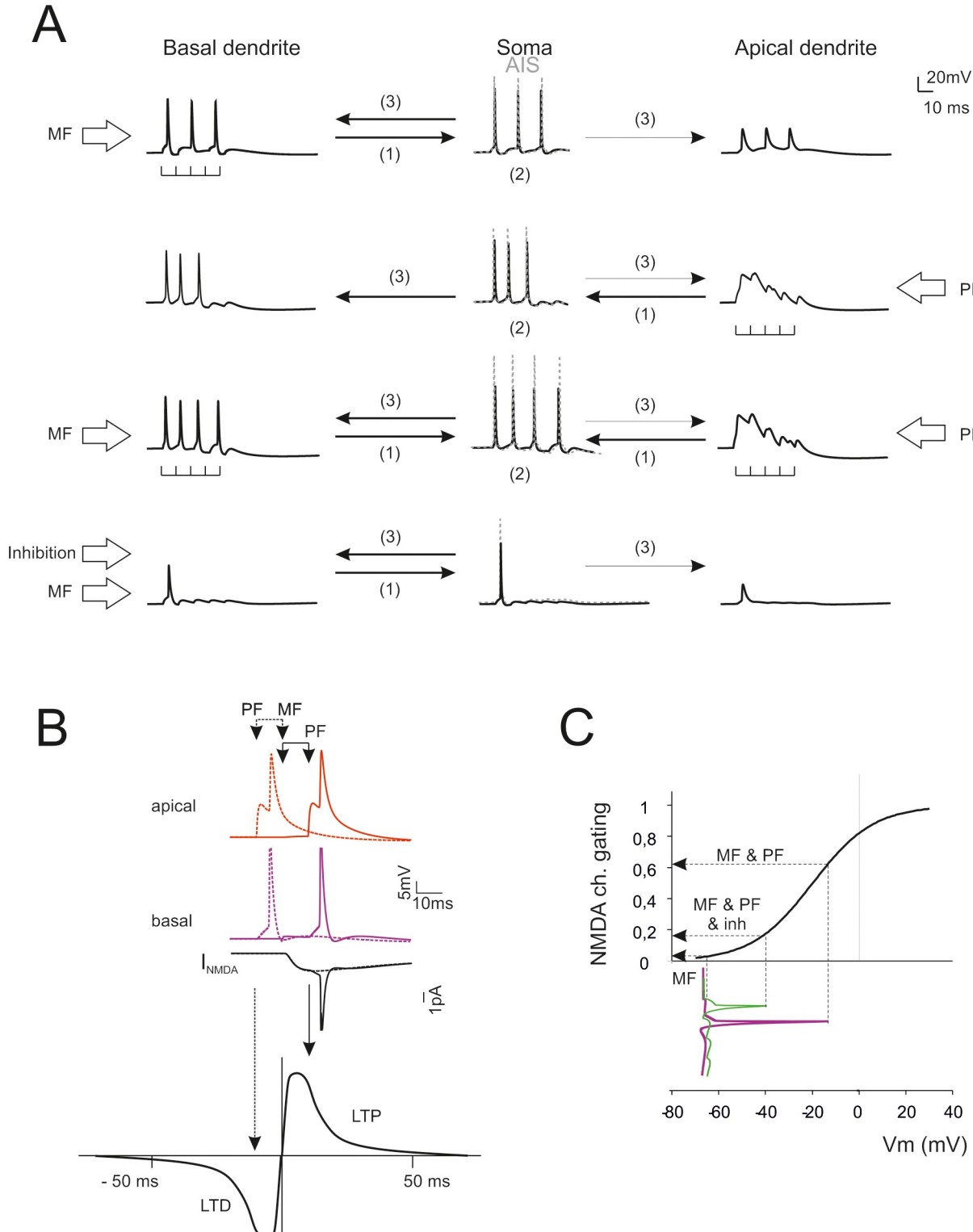

**Fig 9. Predictions of dendritic computation and STDP.** (A) Dendrite–AIS interplay during synaptic transmission. Stimulation of basal dendrites boosts the AIS encoder that sends backpropagating spikes in basal dendrites (only small spikelets are transmitted to the apical dendrites). Stimulation of apical dendrites generates a slow local response boosting the AIS encoder that sends backpropagating spikes in basal dendrites. Conjoint stimulation of apical and basal dendrites reinforces AIS encoding and spike backpropagation in basal dendrites. Activation of inhibitory synapses severely impairs both spike generation and backpropagation. (B) STDP at mossy fiber inputs. Backpropagating spikes

are elicited by PF stimulation either ~10 ms before or ~10 ms after a single synapse activation in the MFs. The NMDA receptor-dependent current ($I_{NMDA}$) generated at the MF synapse is shown in the two cases. The bottom plot shows a theoretical STDP curve (from [34]) showing modest $I_{NMDA}$ changes leading into the LTD region and large $I_{NMDA}$ changes leading into the LTP region. (C) The NMDA receptors operate as coincidence detectors of mossy fiber synaptic transmission and depolarization caused by spike backpropagation. The NMDA channel gating curve (corresponding to voltage-dependent Mg unblock) is intercepted in different points depending on whether spike backpropagation is present (MF & PF), reduced by inhibition (MF & PF & inh), or absent (MF).

dendritic interactions. As said, the importance of ionic channels distribution emerged during dendritic synaptic integration.

## The electrogenic architecture of Golgi cells

As in other central neurons, like Purkinje cells [28,36], inferior olivary cells [37], pyramidal neurons [30,31,38]], in Golgi cells the dendritic calcium channels played a fundamental role. A variety of Ca channels were located in the dendrites [14] and AIS [23,39]. The models showed that, in basal dendrites, Cav2.2 channel activation followed the action potential trajectory generating rapid Ca transients. In the apical dendrites, Cav3.1 channels boosted slow Ca spikes that were enhanced by Cav2.3 channels. It should be noted that the Cav2.2 channels in basal dendrites enhanced pacemaking while the Cav3.1 and Cav2.3 channels of the apical dendrites did not affect it substantially. Moreover, the Cav3.1 channels sustained post-inhibitory rebound bursts. Other Ca channels (Cav2.1, Cav1.x) were not introduced but they may conceivably amplify and modify the local response. Recently, dendritic Ca spikes have been measured in Golgi cells [40] following mossy fiber stimulation and have been related to activation of Cav3.1 and Cav1.x channels and to the induction of long-term synaptic plasticity. The Kca2.2 channels had also fundamental roles: in basal dendrites they caused phase-reset, dumped basal firing frequency and prevented rhythmic low-frequency bursting, while in apical dendrites they limited calcium spike duration. The Kca1.1 channels in soma, dendrites and AIS enhanced spike after-hyperpolarization. In general, the electrogenic architecture of the Golgi cell was such that the basal dendrites were more rapidly interacting with the AIS and the firing mechanism than the apical dendrites, which conveyed to AIS slow currents controlling spike burst generation.

A critical role was played by Nav1.6 channels, which were located at decreasing density in AIS > axon > soma > basal dendrites >> apical dendrites. Due to this differential distribution, fast Na spikes were generated in the AIS [21] and propagated down along the axon. While uncertainty remains about Na channel density in the axon, the assumption of homogeneous channel distribution yields transmission delays compatible with feedback inhibition measured in the granule cell–Golgi cell loop [18]. The spikes also invaded the soma and efficiently backpropagated in basal dendrites but much less so in apical dendrites, which showed heavily filtered spikelets in their terminal part.

A relevant set of ionic channels was located in the AIS, presumably to increase current density and improve control over the firing cycle. These channels controlled high level features of neuron discharge: the HCN channels regulated pacemaker frequency and sags, the M-channels regulated low-frequency resonance, and the LVA channels enhanced burst generation (see [16,17]. Surprisingly the model predicted high local Ca raise in AIS during spikes, similar to Cartwheel neurons of the Cochlear Nucleus and in cerebral cortex L5 Pyramidal cells [22,23]. This effect, which promoted local Kca1.1 and Kca2.2 channel activation, remains to be demonstrated experimentally.

## The synaptic architecture of Golgi cells

A recent estimate of the number of synaptic inputs impinging on Golgi cell dendrites yielded about 400 ascending axon synapse, 4000 parallel fiber synapses, and 40 mossy fiber synapse

per Golgi cell [9]. Here we have partially explored the synaptic input space in order to generate EPSPs and spikes in Golgi cells and investigate the principles of neurotransmission in this neuron. Short-term facilitation at ascending axon and parallel fiber synapses make the Golgi cell poorly sensitive to sparse low-frequency spikes while responding promptly to spike bursts at more than 10 Hz. This high-pass filter at the granule cell—Golgi cell input would prevent, through Golgi cell—granule cell recurrent inhibition, circuit run-away with strong activity in granule cells. Conversely, the mossy fiber synapses, which do not have apparent facilitation or depression during bursts, would allow reliable transmission at all frequencies providing the basis for a linear input-output transformation. Finally, Golgi cell resonance would improve synaptic input tracking in the theta-band, supporting plasticity mechanisms operating on this frequency band.

The effect of synaptic inhibition (illustrated in Fig 7) is clearly to reduce or "dump" the Golgi cell response burst (with an evident effect when inhibition engages more than ~10 synapses). The impact of inhibition on NMDA receptor-dependent long-term synaptic plasticity is illustrated in Fig 9C. In principle, modulating the burst through inhibition would allow to visit a relevant portion of the STDP curve and to modify plasticity accordingly.

### Predictions of dendritic integration

The Golgi cell models provided testable predictions about the interplay of basal and apical dendrites under various patterns of excitation and inhibition [8–10].

First, simulations showed how the dendrites transferred synaptic currents generated by mossy fibers, granule cell ascending axons and parallel fibers. Dendritic filtering reduced and slowed down the AMPA EPSCs on their way to the soma, more effectively in apical than basal dendrites. The NMDA current protracted the time course and enhanced temporal summation at mossy fiber synapses. These simulations confirmed the interpretation of synaptic currents recorded in patch-clamp experiments *in vitro* [9].

Secondly, simulations predicted anew the somatic voltage changes caused by the activation of specific synaptic pathways. All the excitatory synapses showed high efficacy, with just a few active contacts capable of driving a spike, but the mossy fiber and ascending axon synapses proved ~5 times more effective that parallel fiber synapses. Repetitive transmission induced spike bursts followed by phase reset, generating a burst-pause pattern similar to that observed *in vivo* [29]. The inhibitory synapses were strategically placed to shunt currents travelling between the distal part of dendrites and soma and effectively reduced AIS activation as well as backpropagating spikes.

Thirdly, simulations revealed a marked asymmetry in dendritic integration. During repetitive synaptic activation, the basal dendrites showed rapid spike bursts travelling back from the AIS, while the apical dendrites generated slow local depolarizations poorly affected by AIS activity.

The input-output hyperspace was governed by the interaction of basal and apical dendrites (S4 Fig) in a way that resembles cortical pyramidal neurons. The analogy could be extended to dendritic control mechanisms, with the soma and basal dendrites processing Na spikes and apical dendrites processing Ca spikes. It is therefore possible that Golgi cells of cerebellum share common plasticity rules and computational principles with pyramidal neurons [30,31].

### Implications for Golgi cell computation and plasticity

STDP is a form of synaptic plasticity depending on the temporal correlation of two neural events, one leading to synaptic activation of NMDA receptors and the other to depolarizing phenomena (e.g. spike bursts) regulating NMDA channels unblock [30,33,41–43] (S5 Fig).

Indeed, model simulations showed that apical dendrites can regulate the number of AIS spikes that then backpropagate into basal dendrites with consequent regulation of NMDA channel unblock (when the mossy fiber synapses are also active). The availability of NMDA currents, generated in basal dendrites of the Golgi cell model with different synaptic input patterns, allowed to simulate the possible mechanisms of STDP induction in Golgi cells (S5 Fig). The Shouval-Bear-Cooper model of plasticity [42,43] predicted that reliable STDP would occur with an inversion point between LTP and LTD at 0.75 μm $[Ca^{2+}]_i$ and a rate of change of 10. With these parameters, STDP magnitude ranges up to ±50% and increases with the number of active parallel fibers, i.e. with the depolarization conveyed from apical to basal dendrites. The time window of STDP extends over ±100 ms, thereby matching the period of Golgi cell pacing in the theta-band.

In this way, the apical dendrites could integrate multimodal information coming from parallel fibers and then influence the coincidence of spikes with specific synaptic inputs arriving on the basal dendrites. And since many Golgi cells can pulsate together due to gap-junctions and reciprocal inhibition, spike timing would extend over the interneuron network. This property would be important for mossy fiber–Golgi cell STDP, which has been hypothesized [13] to enable the acquisition of incoming mossy fiber patterns in an oscillating network of Golgi cell interneurons. It is worth noticing that the Golgi cells pacemaker oscillation occurs in the theta band, which is the most efficient in generating STDP at the neighboring mossy fiber–granule cell synapses [34] and that granule cells NMDA receptor unblock could be critically controlled by Golgi cells regulating spatiotemporal network coding [7]. Therefore, the Golgi cell electrogenic architecture may play a central role in controlling granular layer network functioning and plasticity as a whole.

## Conclusions

The present simulations, beyond faithfully reproducing the rich pattern of Golgi cell electrophysiological properties recorded *in vitro* and *in vivo*, suggest that Golgi cells operate as oriented coincidence detectors of parallel fiber and mossy fiber activity. The strong coupling of basal dendrites to the AIS encoder makes them easy to invade by backpropagating spikes regulating NMDA channel unblock and STDP induction at the mossy fiber inputs. In turn, apical dendrites can integrate time-dispersed parallel fiber inputs and influence the AIS encoder controlling spike generation. These properties resemble to some extent those reported in pyramidal cells, which also express differentiated ionic mechanisms in their basal and apical dendrites. Oriented dendritic processing could therefore set up a cellular coincidence detector and explain a critical aspect of the learning mechanisms in the cerebellar granular layer. This prediction waits for electrophysiological confirmation using selective optogenetic stimulation of the different synaptic pathways impinging onto the Golgi cells.

## Materials and methods

### Model construction

In this work we have reconstructed and simulated active multicompartmental models of cerebellar Golgi Cells (GoCs) in NEURON [44,45]. The models were based on 8 mouse GoC detailed morphologies. The voltage-dependent ionic channels were distributed according to immunohistochemical, electrophysiological and pharmacological data [8,14]. The gating properties were obtained from previous works [16,17,46–48] taking care of matching as much as possible the GoC genotype. The synaptic distribution and receptor gating properties were also precisely matched to experimental determinations [9,10]. The models were optimized with BluePyOpt [49,50] against features extracted from templates derived from experimental

recordings [15]. This is the first family of GoC models accounting for active excitable properties in the axonal initial segment (AIS), axon, dendrites and soma and for selective activation of different receptor types at distinct synaptic locations.

**Morphology.** Detailed morphologies of 8 GoCs (Neurolucida reconstructions taken from sagittal slice of male and female P23—P29 mice; [19]) were downloaded from NeuroMorpho. org and imported in the NEURON simulator. Four morphologies showed the axon stemming from the soma and four from a basal dendrite. This morphological characteristic is rather common in the brain [51] and was reported also in cerebellar granule cells [52] and in the inferior olivary nucleus. Each morphology was segmented into compartments and the dendrites, axons, soma and axon initial segment (AIS) were identified based on their location and geometrical properties. The distinction between basal and apical dendrites was performed based on their proximity to the soma and Purkinje cell layer [19]. The dendrites contained in the granular layer were classified as basal dendrites, whereas the branches projecting to and located in the molecular layer where classified as apical dendrites. The geometrical properties of model compartments are reported in Supporting Information (S1 Table). It should be noted that the available Golgi cell reconstructions have rather complete dendritic trees but are usually missing part of the axonal plexus [53].

**Passive properties.** The passive properties of the GoC models were set to match passive current transients and spike shape using the following parameters. Axial resistance was set to the standard value of $R_a = 122 \ \Omega \cdot cm$ [28] for all the sections. Membrane capacitance was set to the standard value $C_m = 1 \mu F/cm^2$ in all compartments except in dendrites, were it was set to $C_m = 2.5 \ \mu F/cm^2$ to match spike amplitude. Leakage conductance was set to $G_L = 3 \cdot 10^{-5} \ S/cm^2$ in all compartments except in the axon, where it was set at $G_L = 1 \cdot 10^{-6} \ S/cm^2$ according to recent measurements [54,55]. The reversal potentials were set to +60 mV sodium channels, -80 mV for potassium channels, -20 mV for HCN channels, -70 mV for Cl channels [16,17]. The reversal potential for leakage was adjusted to -55 mV in order to guarantee proper basal firing frequency. The resulting input resistance, calculated with a 10 mV negative step from -70mV, was 339.3 ± 128.3MΩ in agreement with experimental determinations (Forti et al., 2006).

**Membrane mechanisms.** The GoC models, in the absence of more specific information, were endowed with ionic mechanism that were previously validated and used in GrC models [28,55], PC models [27,46,47,56] and Golgi cell models [8,16,17] (see Table 1).

Nav1.6 –The sub-type of GoC sodium channels was determined experimentally [58] but only recently their distribution on the dendritic tree was suggested by electrophysiological experiments [14]. The gating mechanism was taken from [47]. The maximum conductance values were set according to literature, with AIS [8] > soma > basal dendrite > apical dendrite > axon [14].

Kv1.x - The low threshold Kv1.x channels (summarizing properties of Kv1.1 and Kv1.2), in accordance with experiments [62], were placed on the somatic compartment. The gating mechanism was taken from [47].

Kv3.4 –This ionic channel with delayed rectifier properties was restricted to the somatic compartment [63]. The gating mechanism was taken from [47].

Kv4.3 –This ionic channel with A-type properties was placed in the soma. The gating mechanism was taken from a previous GoC model [16,17].

Cav2.2 and Cav2.3 –The high-threshold calcium channels were recently suggested to have distinct locations [14]. Accordingly, the N-type calcium channels (Cav2.2) were placed in basal dendrites. Localization in the soma and AIS was inferred from immunolocalization data [64]. The R-type calcium channels (Cav2.3) were placed in the distal dendrites. The gating mechanism of Cav2.2 was taken from a previous GrC model [48], that of Cav2.3 from [61].

**Table 1. Ionic mechanisms in the GoC model.** The table shows the main properties of ionic channels used in the Golgi cell models. For each ionic channel type, the columns specify the maximum ionic conductance $G_{i-max}$ (value of the model in Fig 1 along with the range of parameters in the 8 models), ionic channels reversal potential $E_{rev}$, the type of ionic channel mathematical representation and the reference from which the ionic channel gating equations have been taken, Markovian or Hodgkin-Huxley (HH).

| Type | Location | $G_{i-max}$ (S/cm²) | $E_{rev}$ (mV) | Type of ionic Channel model | Reference |
|---|---|---|---|---|---|
| **Na CHANNEL** | | | | | |
| **Nav1.6** | Apical Dendrites | 0.0049 (0.0024–0.0068) | 60 | Markovian | [47,57,58] |
| | Basal Dendrites | 0.008 (0.006–0.00135) | | | |
| | Soma | 0.149 (0.143–0.24) | | | |
| | AIS | 0.172 (0.17–0.2) | | | |
| | Axon | 0.011 (0.0072–0.025) | | | |
| **K CHANNEL** | | | | | |
| **Kv1.x** | Soma | 0.005 (0.004–0.007) | -80 | HH | [47] |
| **Kv3.4** | Soma | 0.149 (0.11–0.195) | -80 | HH | [57,58] |
| | Axon | 0.0091 (0.0021–0.010) | | | |
| **Kv4.3** | Soma | 0.004 (0.0035–0.0085) | -80 | HH | [59] |
| **Kv7.x** | AIS | 0.0002 (0.00022–0.0003) | -80 | HH | [16,17,48] |
| **Ca DEPENDENT K CHANNELS** | | | | | |
| **Kca1.1** | Apical Dendrites | 0.01 (0.008–0.014) | | Markovian | [46] |
| | Basal Dendrites | 0.012 (0.009–0.014) | | | |
| | Soma | 0.017 (0.0145–0.019) | | | |
| | AIS | 0.1 (0.09–0.12) | | | |
| **Kca2.2** | Apical Dendrites | 0.0024 (0.0018–0.0045) | -80 | Markovian | [16,17] |
| | Basal Dendrites | 0.016 (0.015–0.019) | | | |
| **Kca3.1** | Soma | 0.0101 (0.09–0.15) | -80 | HH | [60] |
| **CA CHANNEL** | | | | | |
| **Cav2.2** | Basal Dendrites | 0.0013 (0.00125–0.0019) | 137.5 | HH | [48] |
| | Soma | 0.0087 (0.0078–0.0165) | | | |
| | AIS | 0.0059 (0.0045–0.0075) | | | |
| **Cav2.3** | Apical Dendrites | 0.0012 (0.0012–0.0017) | 137.5 | HH | [61] |
| **Cav3.1** | Apical Dendrites | 3.69e-5 (3.6e-5–7.5e-5) | 137.5 | HH | [46] |
| | Soma | 3.40e-5 (3e-5–9.5e-5) | | | |

*(Continued)*

**Table 1.** (Continued)

| Type | Location | $G_{i\text{-}max}$ (S/cm²) | $E_{rev}$ (mV) | Type of ionic Channel model | Reference |
|------|----------|---------------------------|----------------|-----------------------------|-----------|
| **MIXED CATIONIC CHANNEL** | | | | | |
| **HCN1** | AIS | 0.00033 (0.0001–0.00038) | -20 | HH | [16,17] |
| **HCN2** | AIS | 0.0003 (0.0003–0.00035) | -20 | HH | [16,17] |
| **CALCIUM BUFFER (mM)–PUMPS DENSITY** | | **mol/cm2** | | | |
| **Ca Buffer** | Apical Dendrites | 5*10–9 | | Markovian | [46] |
| | Basal Dendrites | 2*10–9 | | | |
| | Soma | 1*10–7 | | | |
| | AIS | 1*10–8 | | | |
| | Axon | 1*10–8 | | | |

Cav3.1 –The presence of low threshold T-type calcium channels (Cav3.1) on the soma was reported by immunohistochemistry [65] and on the apical dendrites was suggested by electrophysiology [14]. Accordingly, we placed Cav3.1 in these locations. The gating mechanism was taken from [46].

KCa1.1, KCa2.2 and KCa3.1 –The big-k calcium-dependent potassium channel (Kca1.1), which can cluster with Cav2.x channels, was placed on the dendritic, somatic and AIS compartments based on immunohistochemical and electrophysiological data [66]. The gating mechanism was taken from [46]. The small-k calcium-dependent potassium channel (KCa2.2) was placed in the dendrites on the basis of electrophysiological recordings showing its importance in limiting calcium-dependent bursting [15] and based on the fact that KCa2.2 doesn't normally coexist with either KCa3.1 or Cav3.1 (see below and the section on results). The gating mechanism was taken from a previous GoC model [16,17], with the caveat that this channel model describes more closely KCa2.2 (SK2) than KCa2.3 (SK3) typical of Golgi cells [67,68]. The middle conductance calcium-dependent potassium channel (KCa3.1) was placed in the soma [69]. The gating mechanism was taken from mitral cells of the olfactory bulb [60].

HCN1 and HCN2 –The H-channels, which were identified electrophysiologically in GoCs [15], have been located in the AIS (Vervaeke, 2012). The gating mechanism was taken from a previous GoC model [16,17].

Kv7.x –The M-current was identified electrophysiologically [15] and was assumed to correspond to Kv7.xchannels [15,64]. Based on anchoring of Kv7.x to Ankyrin-G [70,71], the channels were placed in the AIS using gating mechanisms developed for the GrC [48].

Calcium dynamics–The calcium buffer was taken from a PC model [27,46,56] and modified to contain Parvalbumin [72] and Calmodulin [73], the typical calcium binding proteins of the Golgi cell. The calcium pumps density was increased with respect to default to stabilize spike generation over 2 sec simulations. The density of calcium pumps was set to $2 \cdot 10^{-9}$ mol/cm² for the apical dendrites, $5 \cdot 10^{-9}$ mol/cm² for the basal dendrites, $1 \cdot 10^{-7}$ mol/cm² for the soma and $1 \cdot 10^{-8}$ mol/cm² for the AIS.

**Synaptic mechanisms.** Golgi cells receive synapses from different excitatory pathways (parallel fiber and ascending axon from granule cells, mossy fibers from various brain regions including the DCN; [74]) and from inhibitory neurons. Excitatory synapses have been recently characterized in detail [9]. The parallel fiber synapses were distributed on the apical dendrites, one per compartment. The ascending axon synapses were placed on the basal dendrites, one per section at some distance from the mossy fiber synapses [9]. The mossy fiber synapses were placed on the basal dendrites using sections which were, at least, 20micron away from the soma [75]. The inhibitory synapses on GoCs are still an open issue but recent evidence reports

them both on the apical and basal dendrites [12]. The apical dendrites reportedly receive a low amount of inhibition from molecular layer interneurons and Lugaro cells, while the basal dendrites receive inhibition from other GoCs and the deep cerebellar nuclei (DCN) [76]

The synapses were modelled using dynamic mechanisms for presynaptic neurotransmitter release [24,26], in which key parameters area the recovery and facilitation time constant ($\tau_{REC}$ and $\tau_{FAC}$) and release probability ($p$), and specific kinetic schemes for AMPA, NMDA and GABA receptors. AMPA receptors were placed at mossy fiber and ascending axon synapses on the basal dendrites and at parallel fiber synapses on apical dendrites, NMDA receptors were placed only at mossy fiber synapses on basal dendrites, GABA receptors were placed at Golgi cell inhibitory synapses on both dendrites. Glycinergic synapses [11] and extrasynaptic NMDA receptors [77] were not taken into consideration since their activation would require mechanisms that were not investigated in the present model. The ionic reversal potential was set to 0 mV for AMPA and NMDA receptors and to -70 mV for GABA receptors.

The AMPA receptor kinetic scheme for the parallel fiber and ascending axon synapses was the same as in PCs [27]. The synapses were configured with the following parameters: $p$ = 0.4, $\tau_{REC}$ = 35.1 ms, $\tau_{FAC}$ = 55 ms, and AMPA $G_{max}$ = 1200 pS. This allowed to obtain short-term facilitation with AMPA-EPSCs of ~25pA (at -70 mV) [9] matching EPSCs kinetics recorded from the soma.

The AMPA and NMDA receptor kinetic scheme for the mossy fiber synapses were taken from a granule cell model [55]. The NMDA receptor was modified from the original work with a new kinetic scheme built to reproduce the NR2B subunit-containing receptors [78]. About 10% of the 5.5pA maximum current was converted into calcium influx. The synapses were configured with the following parameters: $p$ = 0.43, $\tau_{REC}$ = 5 ms, $\tau_{FAC)}$ = 8 ms, AMPA $G_{max}$ = 1200pS and NMDA $G_{max}$ = 10000 pS. The simulated EPSC was characterized by a fast AMPA peak (~25pA at -70 mV) followed by a slow NMDA receptor-dependent component and showed temporal summation during repetitive stimulation [9] matching the EPSCs kinetics experimentally recorded from the soma.

The GABA-A receptor kinetic scheme for the inhibitory synapses was taken from recent work that has identify α3β3γ2 GABA subunit containing receptors in the apical and basal dendrites [12,79]. The synapses were configured with the following parameters: $p$ = 0.5, $\tau_{REC}$ = 15 ms, $\tau_{FAC}$ = 4 ms, reversal potential = -70 mV and GABA $G_{max}$ = 2600pS. The simulated IPSC peak (~3.5 pA at -80 mV) matched the EPSCs kinetics experimentally recorded from the soma [10].

## Model optimization

The optimization workflow has been adapted from the one used for the GrC model [28,55]. We used a genetic algorithm to optimize the maximum ionic conductances (the free parameters in the model) for the ionic channels located in all compartments belonging to the same section (apical and basal dendrites, soma, AIS and axon). Optimization occurred against *features* extracted from data reported in [15], which provides highly controlled and reliable whole-cell patch clam recordings of GoC discharge at rest and during current injection. The features used were: action potential width, ISI coefficient of variation (ISI_CV), AHP depth, Action potential height, mean firing frequency, spike count.

The simulations were performed with BluePyopt, using NEURON 7.7 and Python 2.7.15 with fixed time step (0.025ms) [45]. The temperature was set at 32˚ as in slice recordings [15] and all ionic current kinetics were normalized to this value using $Q_{10}$ corrections [28,55]. The stimulation protocol included three positive current injections (0.2, 0.4, 0.6 nA) lasting for 2s.

The optimizations used a population size of 576 individuals and were repeated for 10 generations. The optimizations (running on the Piz Daint—CSCS supercomputer) required 16

nodes (36 cores each) with a computational time ranging from 2:30 to 4:30 hours, depending on the complexity of each morphology.

### Model validation

Model validation consisted in a series of steps, automatically performed with custom-made Python protocols.

The first validation step involved the simulation of each individual of the last generation to assess two fundamental properties: (1) the presence of spontaneous firing and (2) the ability of the models to generate the correct I/O relationship. The first validation criterion required a basal frequency between 2 and 15Hz and an ISI_CV < 0.3 (see Fig 1). The second validation criterion required that the average frequency during the same current steps used for optimization (0.4 nA and 0.6 nA) fell within the experimental range (see Fig 2).

The second validation step assessed the depth of the sag. Each individual was injected with a negative current step (-0.2 nA) lasting for 1000 ms (Fig 3). Validation required that the sag depth fell between 4.4 mV and 10mV [8,15–17]. In order to do so, a specific optimization was run and the successful models were passed to the third validation step. After the firsts and second validation step, the success rate for the eight morphologies was 32 ± 27.1%.

The third validation step assessed the compatibility of the model with experimental pharmacological blockade of Cav2.2, Kca1.1 and Kca2.2 ionic channels [14]. The channels conductance was reduced up to 90%. Models capable of generating bursts, with Cav2.2 partially blocked, were taken in consideration for their response to reductions of Kca2.2 and Kca1.1 conductance. After the third validation step, the valid models were just 10–20% of the initial population.

In aggregate, each model shown in the paper was selected from a large set of possible models (1152). About 60% of the models were discarded because they did not match the typical features of a Golgi cell (i.e. did not show spontaneous firing, an appropriate Input-Output relationship or a sag). Among the models surviving triage, only a part (about 10% of the total) showed correct sensitivity to ionic channel blockers. Out of these ~100 models, one was randomly chosen and shown in the paper. The model parameter space is illustrated in S3 Fig.

### Data analysis

The optimizations results, validations and single simulations where analyzed with custom Python scripts and, for specific cases, with MATLAB 2018a/b provided by the University of Pavia. The morphologies were plotted with Vaa3D [80]. The data used to plot the graphs can be found in S1 Data.

## Supporting information

**S1 Table. Dimension and number of compartments in the Golgi cell model.** The table shows parameters of compartments used in the 8 Golgi cell models taken from NeuroMorpho [19].
(XLSX)

**S1 Data. Source data for the graphs.** The table contains the source data used to plot the graphs for the main text figures.
(XLSX)

**S1 Fig. Electroresponsiveness of the 8 Golgi cell models.** The GoC experimental template is shown to the top along with the step current injection protocol used to elicit the electrical response [15]. The same stimulation protocol was applied to the 8 Golgi cell models. For each model the panels show the morphological reconstruction and the electrical response. Note the

similarity among the models and between them and the experimental case.
(TIF)

**S2 Fig. Variability of ionic conductances in Golgi cell models.** The table shows the maximum ionic channel conductances of 5 randomly chosen individuals for each of the 8 GoC models. The 40 individuals (all validated according to the criteria explained in Materials and Methods) show that the model optimization algorithms thoroughly explored the parameter space providing diverse ionic conductance patterns. In other words, there are different combinations of conductance values that allow to achieve a spike discharge compatible with the experimental templates.
(TIF)

**S3 Fig. Schematics of Golgi cell connectivity.** The figure illustrates the connectivity of GoCs with MFs and GrCs. Inside the glomerulus, a Golgi cell basal dendrite receives excitatory inputs from a MF terminal, while the Golgi cell axon inhibits granule cell dendrites. The granule cells excite the Golgi cell dendrites thought the AAs and PFs. The PF activate only AMPA receptors, while the AA and MF activate both AMPA and NMDA—NR2B-containing receptors. Reciprocal inhibition occurs between the GoC axons and dendrites. Golgi cell; GrC, granule cell; AA, ascending axon; PF, parallel fiber; MF, mossy fiber.
(TIF)

**S4 Fig. The synaptic input-output space of a Golgi cell model.** The synaptic input-output (I/O) space of the Golgi cell model was computed from the response to combined activation of basal and apical dendrites through the corresponding input pathways. The simulations have been repeated using a short train of 5spikes@100Hz on both mossy fibers and parallel fibers and the corresponding output frequency is color-coded. The stars indicate 4 points in the I/O plane that were used to compute STDP, as shown in S5 Fig.
(TIF)

**S5 Fig. Simulation of STDP in the Golgi cell model. (A)** Schematics of the STDP model applied to Golgi cells [42,43]. The coincidence of MF and PF activity regulates NMDA-R activation and channel unblock causing $Ca^{2+}$ entry and a change in $Ca^{2+}$ concentration, $\Delta[Ca^{2+}]_i$. This, in turn, is transformed into STDP by sigmoidal transfer functions accounting for the molecular mechanisms of Ca-dependent plasticity. **(B)** Based on the I/O plots of S4 Fig, our simulations used 11 mossy fiber synapse and an increasing number of parallel fiber synapses (11, 31, 61, 81), both stimulated with a 5spikes@100Hz burst. The $\Delta[Ca^{2+}]_i$ generated by NMDA channels in the corresponding dendritic compartment reflects local Ca regulation, including removal due to diffusion and extrusion but not amplification by local calcium stores. This amplification, e.g. in the neighboring granule cell dendrites, is of about 3 times [81]. Therefore, the $\Delta[Ca^{2+}]_i$ caused by NMDA channel opening was multiplied by 3 times bringing $\Delta[Ca^{2+}]_i$ around the STDP transition point (~0.75 μM). With a rate of change of 10, the model yields a classical STDP curve for the Golgi cell with a gain that depends on the amount of depolarization conveyed by the parallel fibers acting on apical dendrites. **(C)** Dependence of STDP magnitude on the number of active parallel fibers. The gain tends to plateau around ±40%.
(TIF)

**S1 Video. Golgi cell model activation by a mossy fiber burst.** The same GoC model shown in Fig 1 is activated with a burst (5 spikes@100Hz) delivered through the mossy fibers (20 synapses). The model generates a short spike burst followed by a pause that effectively reset the pacemaker cycle. Membrane potential is represented in color code (scale at the top left) on the

model morphology. The plots show membrane potential traces taken in the soma, basal dendrite and apical dendrite.
(MP4)

**S2 Video. Golgi cell model activation by an ascending axon burst.** The same GoC model shown in Fig 1 is activated with a burst (5 spikes@100Hz) delivered through the ascending axons (20 synapses). The model generates a short spike burst followed by a pause that effectively reset the pacemaker cycle. Membrane potential is represented in color code (scale at the top left) on the model morphology. The plots show membrane potential traces taken in the soma, basal dendrite and apical dendrite.
(MP4)

**S3 Video. Golgi cell model activation by a parallel fiber burst.** The same GoC model shown in Fig 1 is activated with a burst (5 spikes@100Hz) delivered through the parallel fibers (89 synapses). The model generates a short spike burst followed by a pause that effectively reset the pacemaker cycle. Membrane potential is represented in color code (scale at the top left) on the model morphology. The plots show membrane potential traces taken in the soma, basal dendrite and apical dendrite.
(MP4)

## Acknowledgments

Special thanks to the HBP Neuroinformatics Platform, HBP Brain Simulation Platform, HBP HPAC Platform for providing access to informatic infrastructures and supercomputing resources.

## Author Contributions

**Conceptualization:** Egidio D'Angelo.

**Formal analysis:** Egidio D'Angelo.

**Funding acquisition:** Egidio D'Angelo.

**Investigation:** Stefano Masoli, Alessandra Ottaviani.

**Methodology:** Stefano Masoli, Stefano Casali.

**Project administration:** Egidio D'Angelo.

**Software:** Stefano Masoli, Alessandra Ottaviani, Stefano Casali.

**Supervision:** Egidio D'Angelo.

**Validation:** Stefano Masoli, Alessandra Ottaviani.

**Visualization:** Stefano Masoli, Alessandra Ottaviani, Egidio D'Angelo.

**Writing – original draft:** Stefano Masoli.

**Writing – review & editing:** Egidio D'Angelo.

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
