## [Decision Letter · Decision Letter 0]

16 Jul 2020

Dear Dr. D'Angelo,

Thank you very much for submitting your manuscript "Cerebellar Golgi cell models predict dendritic processing and mechanisms of synaptic plasticity" for consideration at PLOS Computational Biology. As with all papers reviewed by the journal, your manuscript was reviewed by members of the editorial board and by several independent reviewers. The reviewers appreciated the attention to an important topic. Based on the reviews, we are likely to accept this manuscript for publication, providing that you modify the manuscript according to the review recommendations.

The reviewers identified a number of points that need to be clarified and these concerns should be addressed carefully. In particular, a revised manuscript should either tone down the claims on plasticity results or provide the corresponding simulations.

Sincerely,

Hermann Cuntz

Associate Editor

PLOS Computational Biology

Kim Blackwell

Deputy Editor

PLOS Computational Biology

[LINK]

The reviewers identified a number of points that need to be clarified and these concerns should be addressed carefully. In particular, a revised manuscript should either tone down the claims on plasticity results or provide the corresponding simulations.

Reviewer's Responses to Questions

**Comments to the Authors:**

Reviewer #1: The manuscript “Cerebellar Golgi cell models predict dendritic processing and mechanisms of synaptic plasticity” from Masoli et al is a solid description of a new model of GoCs that recapitulate many of the intrinsic properties known to exist in cerebellar GoCs. They use a machine learning method -- ‘genetic optimization’ -- to find channel conductance values that produce known electrophysiological signatures. They go on to do experiments with the model, leaving out various conductances, to identify KCa conductances and a link with Ca2.2 as critical in steady pacemaking, as their removal promotes a bursting phenotype. Synaptic inputs to both apical and basal dendrites elicit a ‘burst’ of spikes and through asymmetry in action potential back propagation speed may support coincidence detection and subsequent spike time dependent plasticity. The manuscript was very clear, well written and may become a definitive model of Golgi cells.

Below I provide minor suggestions for improvement.

Main critiques.

1. The optimization process used trains on known intrinsic properties of Golgi cells. I would imagine that with the number of currents the authors use that multiple solutions are found, producing a solution ‘space’. Rather the findings are more rigid, with one answer per morphology. Can the authors please address this or provide a description of the parameter spaces found that recapitulate known Golgi cell intrinsic properties?

2. The model of the synaptic inputs seems rather limited. What are the studies that support the numbers chosen for mossy fibers (20), and parallel fibers (89 synapses)? These seem very specific and not necessarily correct. Relatedly, I found the focus on short-term synaptic plasticity to be interesting but not necessarily complete for the potential roles of Golgi cells in the circuit. The authors might also comment on the ability of the Golgi cells rate to track synaptic inputs to their basal dendrites, apical dendrites or both in ways that could enhance computation within the granule cell layer – possibly expanding the description of data in Figure 7.

3. The authors should mention that the Golgi cell reconstructions from slices are necessarily incomplete, so although these reconstructions are valuable for these purposes they are not as large as Golgi cells actually are.

4. Ln 318-320, the authors could expand on the observation that inhibition “dumps the burst” and influences LTP. I found the comment terse and could be expanded upon. Do the authors mean “prevents the burst” and if so, analyses on to what extent the spike rate is reduced to prevent LTP are warranted.

Minor

1. The final sentence of the Methods seems to be an incomplete though or just a punctuation typo.

2. I didn’t follow the language in the methods, used twice, that “This allowed to prevent” ln 581 or “This allowed prevent” ln 575. What is “this” and what is allowed to prevent in these sentences? Prevented?

Reviewer #2: The article represents a valuable contribution to computational modeling of the cerebellum and proposes an interesting mechanisms of cerebellar plasticity. It is well argued, well written and well researched.

The main conclusion, namely that the STDP in basal dendrites seems to be controlled by parallel fiber information, is interesting and well argued via the NMDA mechanism. However, there is no ‘plasticity model’ to speak of. The argument for the presence of STDP is indirect. For instance, there is no model of the rate of synaptic change, and no discussion about the interaction between intrinsic and synaptic mechanisms. This would enrich the scope of the article, and i recommend including references and a concise discussion.

The topic and problem are well introduced and the article goes straight to the point. The cerebellar loops are outlined and it is clear why it is important to look at “the fundamental issues that remain unexplored”. The article includes a substantially updated Golgi cell model which is an important piece of the cerebellar puzzle. The chosen ionic channels and their distributions along the cell are well justified and the need for the model is explained.

The inclusion of the model’s history is valuable, and the improvements from morphological and immunohistochemical data are compelling. Model optimization and validation are straight forward and well referenced. The model follows the electrophysiological and immunohistochemical data and is shown to respond well to different somatic current injections (eliciting sag, rebound bursting, phase-reset, etc).

STDP diagram is the standard one and is shown in Fig 9 how the changes in I_NMDA lead to LTD or LTP regions. These are referenced [13 and 31] but adding a short exposition of the conclusions of those articles will make the article more self-contained. It would also be very important to explain what are the relevant scales of change.

To my mind, the most important omission in the manuscript are the quantitative aspects of STDP. What is the expected maximum modulation of synaptic conductance via the mechanisms assumed? What is the rate of change? What is the Calcium level that switches from LTP and LTD? Going forward it is important to understand the balance between LTP and LTD given expected statistics of synaptic inputs.

For completeness of exposition, it would be relevant to include diagrams of the molecular mechanisms thought to underly bi-directional plasticity. While the mechanisms for LTP are clearly referenced, the detailed mechanisms for LTD are not as clear.

— Suggestions for improvement:

- a simple sketch of the loops modeled would help readers that are not familiar with the cerebellum.

- more self-contained exposition (predicted by theory: what is the prediction and why?)

- discussion of molecular mechanisms involved

- discussion of the dynamic range and rate of STDP

**Have all data underlying the figures and results presented in the manuscript been provided?**

Reviewer #1: Yes

Reviewer #2: **No: **Though the article says the models will be available in the HBP as a live article, no access has been provided for this reviewer (AFAICD).

PLOS authors have the option to publish the peer review history of their article (what does this mean?). If published, this will include your full peer review and any attached files.

Reviewer #1: No

Reviewer #2: No
---

## [Decision Letter · Decision Letter 1]

13 Nov 2020

Dear Dr. D'Angelo,

We are pleased to inform you that your manuscript 'Cerebellar Golgi cell models predict dendritic processing and mechanisms of synaptic plasticity' has been provisionally accepted for publication in PLOS Computational Biology.

Please address the final request by Reviewer 2 to avoid confusion when others use your model.

Best regards,

Hermann Cuntz

Associate Editor

PLOS Computational Biology

Kim Blackwell

Deputy Editor

PLOS Computational Biology

Reviewer's Responses to Questions

**Comments to the Authors:**

Reviewer #1: The authors have addressed my comments and I have no further concerns, except that the figures on my version of the manuscript were of poor resolution and will need to be fixed.

Reviewer #2: Nice work. Your article contains a rich modeling effort and neatly detailed description of channel interactions in the Golgi cell dendrites. On its merits, it is staged to become a new reference Golgi cell models.

Fig 9.b

There is one aspect of the discussion that i would like to bring to your attention. In the discussion about Calcium channel subtypes from line 371 onwards, the authors employ a naming convention (T- Type, R type...) that is at odds with the one used to described the model (Cav3.1, Cav2.2,...). I would starkly recommend making sure they are consistent (e.g., using both when the model is described).

**Have all data underlying the figures and results presented in the manuscript been provided?**

Reviewer #1: Yes

Reviewer #2: Yes

PLOS authors have the option to publish the peer review history of their article (what does this mean?). If published, this will include your full peer review and any attached files.

Reviewer #1: No

Reviewer #2: **Yes: **Mario Negrello

---

## [Editor Report · Acceptance letter]

23 Dec 2020

PCOMPBIOL-D-20-00755R1 

Cerebellar Golgi cell models predict dendritic processing and mechanisms of synaptic plasticity

Dear Dr D'Angelo,

I am pleased to inform you that your manuscript has been formally accepted for publication in PLOS Computational Biology. Your manuscript is now with our production department and you will be notified of the publication date in due course.

With kind regards,

Livia Horvath
